

# Difference spectrum fitting of the ion-neutral collision frequency from dual-frequency EISCAT measurements

Florian Günzkofer[1], Gunter Stober[2], Dimitry Pokhotelov[3], Yasunobu Miyoshi[4], and Claudia Borries[1]

[1]Institute for Solar-Terrestrial Physics, German Aerospace Center (DLR), Neustrelitz, Germany
[2]Institute of Applied Physics & Oeschger Center for Climate Change Research, Microwave Physics, University of Bern, Bern, Switzerland
[3]Institute of Physics, University of Greifswald, Greifswald, Germany
[4]Department of Earth and Planetary Sciences, Kyushu University, Fukuoka, Japan

**Correspondence:** Florian Günzkofer (florian.guenzkofer@dlr.de)

**Abstract.** Plasma-neutral coupling in the mesosphere/lower thermosphere strongly depends on the ion-neutral collision frequency across that region. Most commonly, the collision frequency profile is calculated from the climatologies of atmospheric models. However, previous measurements indicated that the collision frequency can deviate notably from the climatological average. Direct measurement of the ion-neutral collision frequency with multifrequency Incoherent Scatter Radar (ISR) measurements has been discussed before, though actual measurements have been rare. The previously applied multifrequency analysis method requires a special simultaneous fit of the two Incoherent Scatter spectra which is not possible with standard ISR analysis software. The *difference spectrum* method allows inferring ion-neutral collision frequency profiles from multifrequency ISR measurements based on standard Incoherent Scatter analysis software such as the GUISDAP software package. In this work, we present the first results by applying the *difference spectrum* method. Ion-neutral collision frequency profiles obtained from several multifrequency EISCAT ISR campaigns are presented. The profiles obtained with the *difference spectrum* method are compared to previous collision frequency measurements, both from multifrequency ISR and other measurements, as well as results from empirical and comprehensive atmosphere models. Ion-neutral collision frequency measurements can be applied to improve first-principle ionospheric models.

## 1 Introduction

The magnetospheric current system is closed in the ionospheric dynamo region at approximately $80-130$ km altitude due to the maxima of transverse conductivities at these altitudes (Baumjohann and Treumann, 1996). These maxima occur due to the special relations of ion/electron-neutral collision frequencies ($\nu_{in}$, $\nu_{en}$) and ion/electron gyrofrequencies ($\omega_i$, $\omega_e$). In the dynamo region, electrons are coupled to the Earth's magnetic field lines ($\nu_{en} \ll \omega_e$) whereas ions are decoupled due to frequent collisions with neutral particles ($\nu_{in} \gg \omega_i$) (Brekke et al., 1974). Due to the transition from a collisional to a collisionless plasma, the dynamo region is also often referred to as the ionospheric transition region and we will use both terms synonymously.

Knowledge of the vertical collision frequency profiles is essential for understanding and predicting ionospheric conductivities





and their impact on the atmosphere-ionosphere coupling, e.g. due to Joule dissipation of Pedersen currents. The impact of ion-neutral collisions on the diffusion of meteor trails makes direct collision frequency measurements important for the analysis of

meteor radar data as well (Stober et al., 2023). Also, neutral wind measurements with Incoherent Scatter Radars (ISR) require *a priori* knowledge of the collision frequencies in the dynamo region (e.g., Brekke et al., 1973; Nozawa et al., 2010; Günzkofer et al., 2022). $\nu_{in}$ and $\nu_{en}$ can be calculated leveraging simplified and idealized relations derived by Chapman (1956) for the ion-neutral collision frequency

$$\nu_{in} = 2.6 \cdot 10^{-9} \cdot (n_n + n_i) \cdot A^{-0.5}, \tag{1}$$

and by Nicolet (1953) for the electron-neutral collision frequency

$$\nu_{en} = 5.4 \cdot 10^{-10} \cdot n_n \cdot T_e^{0.5}. \tag{2}$$

In Eq. 1 and 2, $n_n$ and $n_i$ are neutral and ion volume densities per cubic centimeter, $A$ is the mean atomic mass number and $T_e$ is the electron temperature (Kelly, 2009). However, the neutral density $n_n$ is often taken from empirical atmosphere models such as NRLMSISE-00 (Picone et al., 2002) which introduce a major source of uncertainty when estimating the colli-

sion frequencies from the above equations. The relations in Eq. 1 and 2 are derived assuming ion-neutral and electron-neutral collisions as elastic rigid-sphere collision (Nicolet, 1953; Chapman, 1956).

There are multiple approaches for direct or indirect measurements of collision frequencies in the ionosphere.

It has been shown that it is possible to infer the ion-neutral collision frequency utilizing line-of-sight ion velocity $v_i$ measurements from ISRs (Nygren et al., 1987). Such an indirect measurement method, however, requires specific conditions and assumptions and is therefore only applicable above $\sim 106$ km altitude and for electric fields $E \gtrsim 20$ mVm$^{-1}$ (for a more

detailed discussion of this method, see Oyama et al., 2012, and references there within). The ion-neutral collision frequency can also be directly inferred from ISR measurements due to its impact on the incoherent scatter spectrum (e.g., Dougherty and Farley, 1963; Farley, 1966; Grassmann, 1993a). However, the shape of the incoherent scatter spectrum is ambiguous towards changes in the collision frequency and changes in ion and electron temperatures $T_i$ and $T_e$. The ambiguity can be overcome

by assuming $T_e = T_i$, however, this assumption has to be considered carefully. A summary and discussion of multiple studies following this approach with the EISCAT ISR can be found in Nygrén (1996). In these studies, it was generally assumed that at high latitudes, $T_e = T_i$ is valid for low geomagnetic activity conditions and below $\sim 110$ km altitude.

The ion-neutral collision frequency can also be inferred from two Incoherent Scatter spectra obtained with simultaneous measurements of two ISRs with well-separated transmitter frequencies. This is possible at the EISCAT site in Tromsø, Norway,

where two ISRs with transmitter frequencies of 929 MHz and 224 MHz are operated. Many methods to analyze EISCAT dual-frequency measurements to obtain ion-neutral collision frequencies have been suggested (Grassmann, 1993b). However, actual dual-frequency measurements have been scarce and only one study performed an analysis to derive collision frequencies (Nicolls et al., 2014). It should be noted that the EISCAT ISR systems are the only ones capable to perform dual-frequency





measurements. In their study, Nicolls et al. (2014) followed an approach suggested by Grassmann (1993b) that requires the simultaneous fitting of both measured spectra to obtain the optimum parameters from the combined fit. This method requires a customized ISR spectrum fitting algorithm and therefore cannot be applied within the standard EISCAT analysis software (GUISDAP). An alternative method suggested by Grassmann (1993b) is the so-called *difference spectrum* fitting, which combines the spectra after the standard ISR analysis.

In this work, the applicability of the *difference spectrum* fitting for the measurement of ion-neutral collision frequencies using the two EISCAT ISR systems (UHF and VHF) is demonstrated. We compare the obtained profiles to the results from Nicolls et al. (2014) as well as to other collision frequency measurement methods. The vertical profile of neutral particle density can be inferred from the measured ion-neutral collision frequency, either by applying Eq. 1 or any other collision frequency relation, though most require certain assumptions on the atmospheric composition across the ionospheric dynamo region. The neutral density profiles can be partially validated by meteor radar measurements in the mesosphere-lower thermosphere (MLT) region (Stober et al., 2012; Stober et al., 2014; Dawkins et al., 2023) or by leveraging occultation measurements from satellites of x-ray sources such as the Crab Nebula (Katsuda et al., 2023). Since measurements of atmospheric densities in the ionospheric dynamo region are very rare, dual-frequency ISR measurements allow us to obtain valuable information required for the validation of atmosphere models. The results of our analysis will be compared to several atmosphere models.

## 2 Instruments and models

### 2.1 EISCAT UHF and VHF radar

The EISCAT Scientific Association operates an Ultra High Frequency (UHF) and a Very High Frequency (VHF) ISR with frequencies of 929 MHz and 224 MHz near Tromsø, Norway (69.6° N, 19.2° E) (Folkestad et al., 1983). The UHF transmitter operates with a power of about $1.5 - 2$ MW. The dish used for transmitting and receiving is 32 m in diameter. The VHF transmitter has a peak power of about 1.5 MW and the co-located VHF receiver antenna consists of four rectangular (30 m x 40 m) dishes. To perform the dual-frequency analysis, both systems have to be operated at the same time in the same radar mode. A summary of all EISCAT experimental modes can be found in Tjulin (2021). In this work, multiple EISCAT campaigns are analyzed.

**August 2013 and September 2021 campaigns**

Nicolls et al. (2014) planned and analyzed the EISCAT campaign on 29 August 2013 from 7 - 11 UTC. The measurements were conducted with both ISRs operated in the *beata* radar mode, pointed to the geographical North with an elevation of 45°. This experiment mode allows measurements from about 80 to 500 km altitude with a vertical resolution of $5 - 10$ km in the transition region.

On 27 September 2021 from 8 - 12 UTC, a dual-frequency EISCAT campaign was conducted leveraging the same radar mode and geometry.



## October 2022 campaign

On 13 October 2022 from 8 - 13 UTC, a dual-frequency EISCAT campaign was conducted in the *manda zenith* experiment mode, also known as the EISCAT *Common Programme (CP) 6*. As the name suggests, the radar is pointed to the local zenith at 90° elevation. This mode allows measurements between about 70 and 200 km with a vertical resolution of about $0.4 - 10$ km in the transition region.

### 2.2 NRLMSIS

Since measurements of ion-neutral collision frequencies are rarely available, climatological profiles from empirical atmosphere models often have to be assumed. One application of such profiles is the derivation of neutral winds from ISR ion velocity measurements (e.g., Nozawa et al., 2010; Günzkofer et al., 2022). The Mass Spectrometer and Incoherent Scatter Radar (MSIS) model (Hedin, 1991) is available in many different versions and is one of the most commonly used empirical atmosphere models. In this paper, we will use empirical atmosphere models both for comparison to measurement data and to obtain profiles of neutral atmosphere parameters that are not available from measurements. Two MSIS versions are applied, NRLMSISE-00 (Picone et al., 2002) and NRLMSIS 2.0 (Emmert et al., 2021).

### 2.3 GAIA

The Ground-to-Topside Model of Atmosphere and Ionosphere for Aeronomy (GAIA) is a global circulation model giving neutral dynamics for all altitudes from the ground up to $\sim 600$ km (Jin et al., 2012). GAIA output has been compared and verified with experimental data from numerous different apparatuses for time spans up to several decades. A comparison to meteor radar wind climatologies at the MLT can be found in Stober et al. (2021). GAIA simulations are nudged up to $\sim 30$ km altitude to the Japanese Re-Analysis data (JRA-25/55 Kobayashi et al., 2015). In this study, we use data from GAIA on a grid with a resolution of 1° in latitude and 2.5° in longitude. GAIA output is available with a vertical resolution of $1/5$ of the scale height.

## 3 *Difference spectrum* fitting of the ion-neutral collision frequency

In this section, we describe the *difference spectrum* fitting method equivalent to Grassmann (1993b), though we changed the notation of some variables. Furthermore, results from multiple EISCAT campaigns are presented. As mentioned in Section 1, the main advantage of this method over other approaches described in Grassmann (1993b) is that it can be applied on top of already existing ISR analysis. It, therefore, does not require a detailed knowledge of the ISR spectrum fitting process. For all EISCAT data presented in this paper, the fitting of ISR spectra is done with Version 9.2 of the Grand Unified Incoherent Scatter Design and Analysis Package (GUISDAP) (Lehtinen and Huuskonen, 1996). As *a priori* guesses for the spectra fits, GUISDAP utilizes the climatology models MSIS and IRI. For a weak signal-to-noise ratio, the fitting returns the *a priori* climatology parameter profiles. However, for a sufficiently good signal quality, the choice of *a priori* climatology has no impact





on the fitted parameters.

In the following, we will distinguish between measured spectra $S\left(\omega_x + \delta\omega\right)$ and theoretical spectra $s\left(\omega_x + \delta\omega, N_e, T_i, T_e, \nu_{in}, v_i\right)$ with the transmitter frequency $\omega_x$. During the fitting process, the measured incoherent scatter spectra $S\left(\omega_{VHF} + \delta\omega\right)$ and $S\left(\omega_{UHF} + \delta\omega\right)$ are saved. The measured VHF spectrum can be scaled to UHF frequencies knowing the ratio $\xi = \omega_{UHF}/\omega_{VHF} \approx 4.15$ for the EISCAT systems. According to Grassmann (1993b), the scaled VHF spectrum $\tilde{S}\left(\omega_{UHF} + \delta\omega\right)$ is equivalent to a UHF incoherent scatter spectrum for the scaled parameters $\xi^2 \cdot N_e$ and $\xi \cdot \nu_{in}$, meaning

$$\xi^2 \cdot S\left(\omega_{VHF} + \delta\omega\right) = \tilde{S}\left(\omega_{UHF} + \delta\omega\right) = s\left(\omega_{UHF} + \delta\omega, \xi^2 \cdot N_e, T_i, T_e, \xi \cdot \nu_{in}, v_i\right). \tag{3}$$

The UHF-spectrum and scaled VHF-spectrum can be combined into a single *difference spectrum*

$$D\left(\omega_{UHF} + \delta\omega\right) = S\left(\omega_{UHF} + \delta\omega\right) - \beta \cdot \tilde{S}\left(\omega_{UHF} + \delta\omega\right), \tag{4}$$

where the additional parameter $\beta$ is included to account for technical differences between the UHF and VHF radars. As described in Grassmann (1993b), $\beta$ is determined at sufficiently high altitudes where $\nu_{in} = 0$ and therefore $D\left(\omega_{UHF} + \delta\omega\right) = 0$ can be assumed.

Figure 1 compares the measured UHF spectrum (top left) and the VHF spectrum after scaling (top right). It can be seen that the spectral intensity maximizes at approximately $200 - 300$ km altitude. $\beta$ is therefore determined at about 260 km where the plasma can be assumed as collisionless. Figure 1 (middle) shows the UHF and VHF spectra at three selected altitudes: at 260 km, where $\beta$ is determined, and at two altitudes in the upper and lower transition region at 160 km and 100 km. At 260 km, both spectra exhibit the typical ISR double-peak shape. As described by Grassmann (1993b), the double-peak shape disappears with increasing $\nu_{in}$, first for the VHF spectrum and at lower altitudes for the UHF spectrum as well. In Fig. 1 (bottom), the UHF and scaled VHF spectrum at 260 km altitude are shown for a 10 min interval with the statistical uncertainty throughout that period. Whether the uncertainties are caused by a variation of the ionospheric plasma parameters within this interval or the result of uncertainties of the Incoherent Scatter measurements can not be conclusively determined. The measured *difference spectrum* is calculated according to Eq. 4 and fitted to the theoretical *difference spectrum* function

$$d\left(\omega + \delta\omega, N_e, T_i, T_e, \nu_{in}, v_i\right) = s\left(\omega_{UHF} + \delta\omega, N_e, T_i, T_e, \nu_{in}, v_i\right) - \beta \cdot s\left(\omega_{UHF} + \delta\omega, \xi^2 \cdot N_e, T_i, T_e, \xi \cdot \nu_{in}, v_i\right). \tag{5}$$

It can be seen from Eq. 3 that the scaled VHF spectrum corresponds to a UHF spectrum with scaled collision frequency $\xi \cdot \nu_{in}$. The parameters $T_i$ and $T_e$ are not affected by the frequency scaling. Therefore, the *difference spectrum* in Eq. 5 allows us to infer $\nu_{in}$, $T_i$, and $T_e$ without ambiguity or any further assumptions. Figure 2 shows the vertical profile of the ion-neutral collision frequency at $90 - 150$ km altitude for 27 September 2021 8 - 12 UTC.

The fitted ion-collision frequency profile shows reasonable values across the whole ionospheric transition region. The error-bars shown in Figure 2 mark the upper and lower quartile of the geophysical variation during the four hours of measurement.





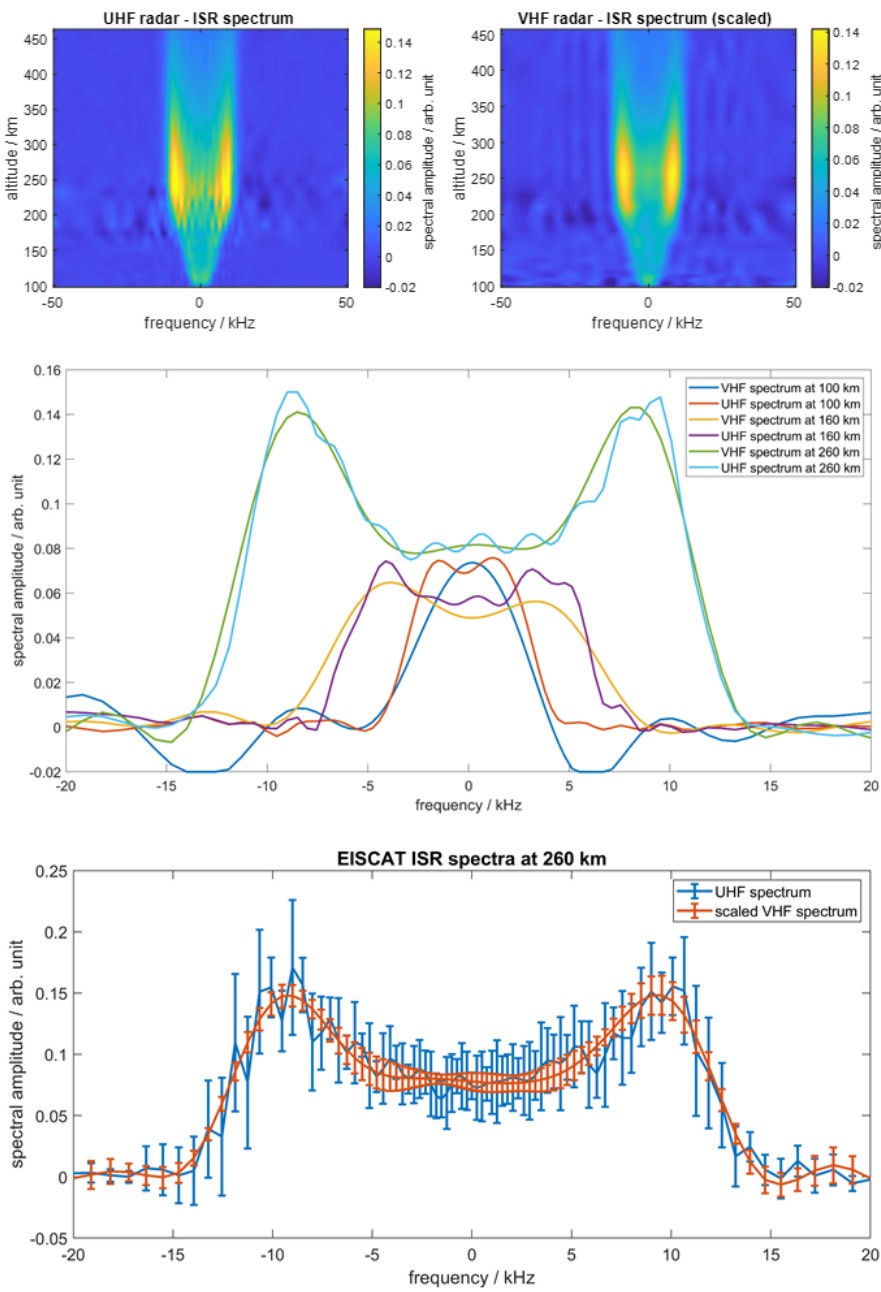

**Figure 1.** Top: EISCAT UHF (left) and scaled VHF (right) spectra measured on 27 September 2021. Middle: Comparison of spectra at 260 km, 160 km, and 100 km altitude. It can be seen how the incoherent scatter spectrum changes shape with increasing collisionality. Bottom: The UHF and scaled VHF at 260 km altitude with statistical uncertainties over a range of 10 min.



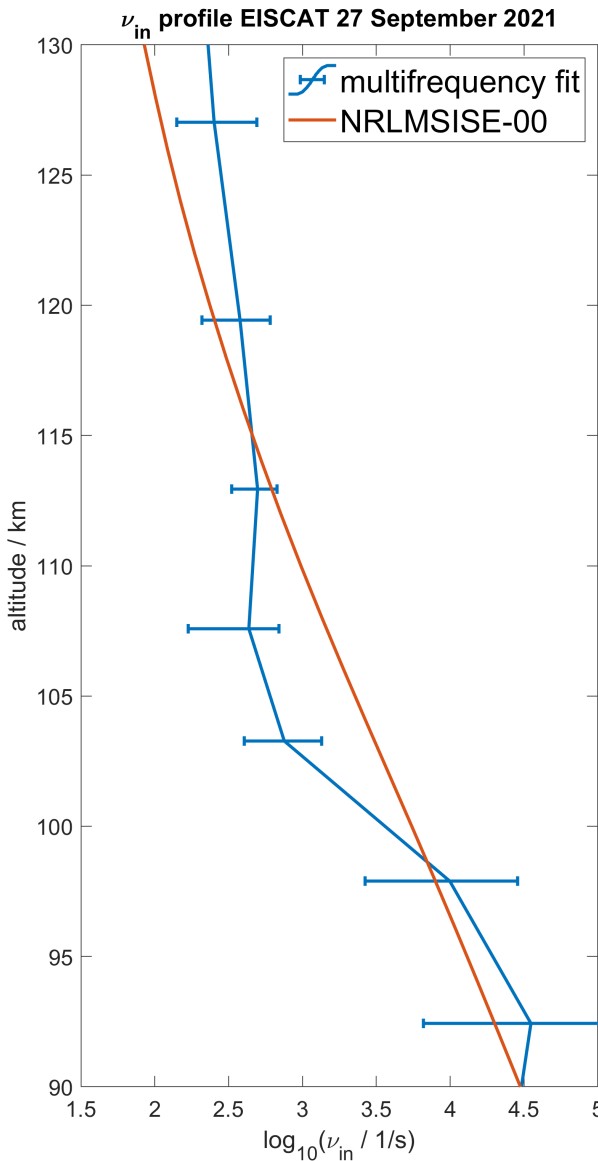

**Figure 2.** Median vertical profile of $\nu_{in}$ on 27 September 2021 at 08 - 12 UTC. One profile fit is performed for every 60 s and the error bars mark the statistical interquartile range. For comparison, a climatological median profile is calculated from the NRLMSISE-00 model.

The uncertainties of the ISR spectra are not obtained during the GUISDAP fitting process. However, it can be assumed that for a large enough signal-to-noise ratio the geophysical variation exceeds the effects of the ISR spectrum uncertainty. Below ~ 120 km, the fitted profile oscillates around the climatological mean calculated with Eq. 1 from the NRLMSISE-00 neutral density. At altitudes above 120 km, the fitting method shows a general tendency to larger collision frequencies in comparison with the NRLMSISE-00 model.



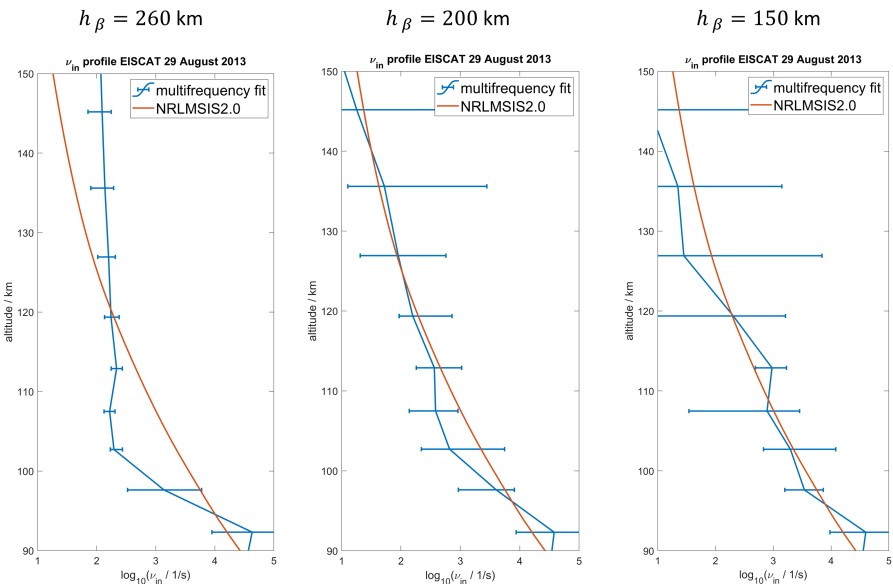

**Figure 3.** $\nu_{in}$ profiles for three different $h_\beta$. The $\beta$ parameter was determined at 260 km (left), 200 km (middle) and 150 km (right) altitude.

As mentioned earlier, there is only one previous multifrequency ISR experiment. The EISCAT campaign from 27 September
2021 was run in the same radar mode as the experiment on 29 August 2013. This allows us to directly apply the developed
*difference spectrum* fitting method on these measurements and compare them to the results shown in Nicolls et al. (2014). How-
ever, the fitted profiles for this campaign show a strong dependence on which altitude $h_\beta$ is used to determine the $\beta$ parameter.
The fitted $\nu_{in}$ profiles for three different altitudes $h_\beta$ are shown in Fig. 3 for 20 August 2013 7 - 11 UTC.

The $h_\beta = 260$ km profile deviates strongly from the climatological NRLMSIS profile and shows a nearly constant collision
frequency at $\sim 110 - 150$ km altitude. This is unexpected and has to be discussed carefully. The two profiles for $h_\beta = 200$
km and $h_\beta = 150$ km show more resemblance to the climatological average, though the propagated uncertainties are larger
compared to the first profile. The $h_\beta = 150$ km profile is also very similar to the profile shown in Nicolls et al. (2014). The
vertical profiles of the $\beta$ parameter versus the altitude where it is determined are shown in Fig. 4 for the two campaigns on 29
August 2013 and 27 September 2021.

It can be seen that the $\beta$ parameter has changed in the time between the two campaigns. The variation with altitude is more
pronounced for the August 2013 campaign with a altitude change rate of $1.22 \cdot 10^{-4}$ km$^{-1}$. For the September 2021 campaign,
the gradient of the $\beta$ profile is $4.78 \cdot 10^{-5}$ km$^{-1}$. This might explain the distinct changes of the August 2013 $\nu_{in}$ profile shown
in Fig. 3. However, what causes these changes of $\beta$ with altitude remains to be discussed.

The third dual-frequency EISCAT campaign was conducted on 13 October 2022. For this campaign, a different radar mode
was applied which enables a better vertical resolution in the MLT region in exchange for a reduced absolute vertical coverage.
Figure 5 shows the $\nu_{in}$ profile obtained from simultaneous *manda zenith* measurements with the EISCAT UHF and VHF
radars at $80 - 100$ km altitude. A climatology profile from the NRLMSISE-00 model is shown, as well as the *a priori* collision



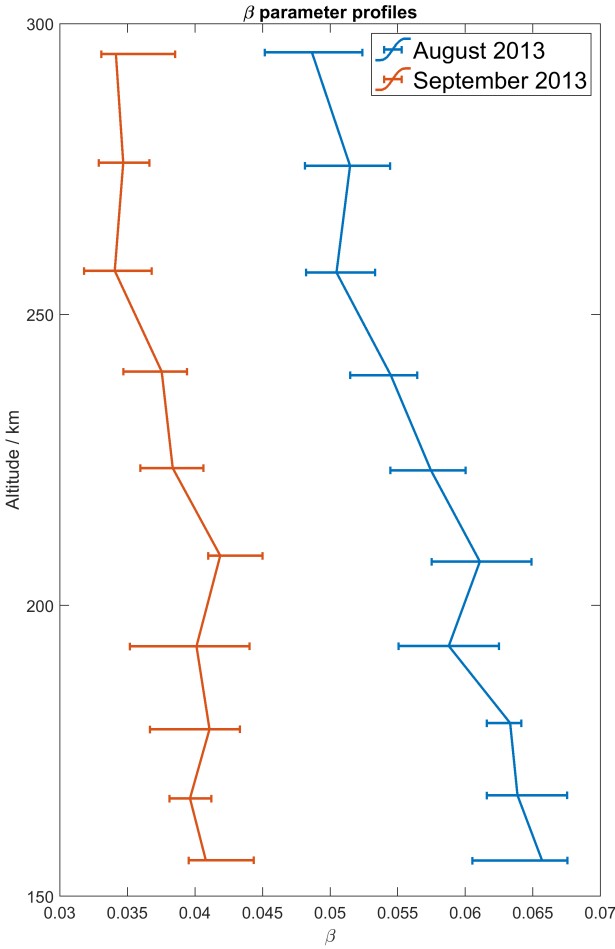

**Figure 4.** Profiles of the $\beta$ parameter vs the altitude where it has been determined for the EISCAT campaigns on 29 August 2013 and 27 September 2021.

frequency profile applied during the *difference spectrum* fit. The *a priori* profile is obtained from the $\nu_{in}$ values given by the single-frequency GUISDAP analysis of the UHF measurements. The single-frequency result is close to a climatological profile
with slightly lower values than the NRLMSISE-00 profile.

It can be seen in Fig. 5 that the dual-frequency fit has almost no measurement response at altitudes $\lesssim 85$ km where the fitted profile is identical to the *a priori* profile. Thus, at those altitudes, there is not enough signal-to-noise ratio left to drive the profile away from the *a priori*.





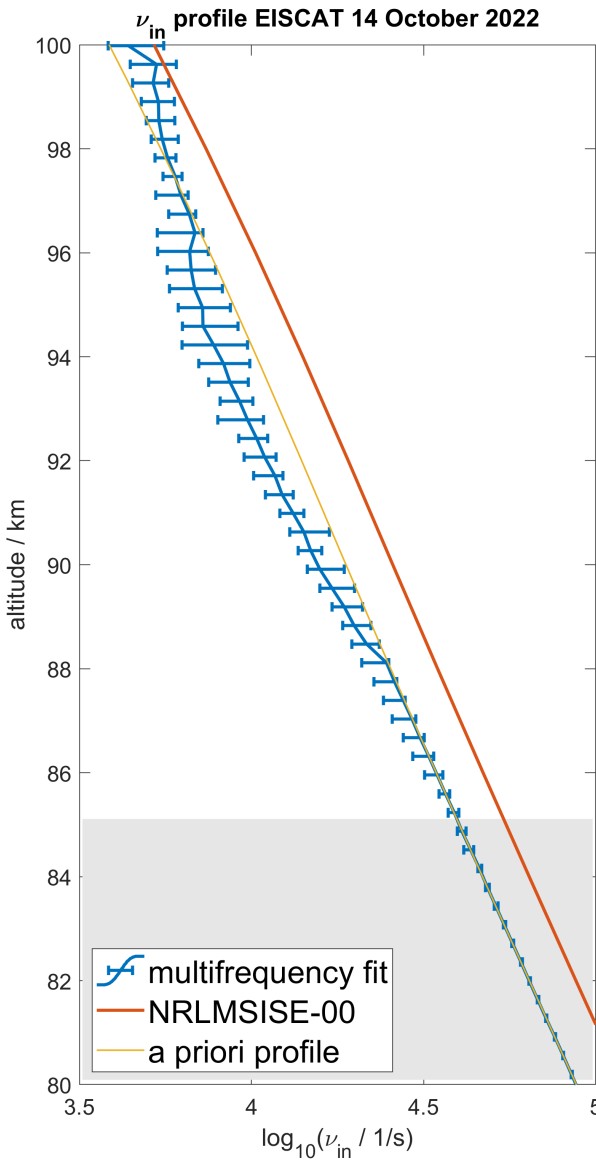

**Figure 5.** High altitude resolution $\nu_{in}$ profile from EISCAT *manda zenith* measurements. The grey area marks the altitude where the dual-frequency fit did not converge.

## 4 Neutral density measurements and comparison to atmospheric models

Equation 1 describes how the ion-neutral collision frequency can be calculated from the neutral particle density $n_n$ and the mean ion mass $A$ assuming the mean ion mass equals the mean neutral atom/molecule mass. Equation 1 assumes ion-neutral collisions as rigid-sphere collisions (Chapman, 1956) and allows calculating the neutral particle densities from the measured



ion-neutral collision frequency profiles. Instead of rigid-sphere collisions, the ion-neutral collision frequency can be calculated assuming Maxwell collisions (Dalgarno et al., 1958). When calculating the neutral particle density with this method, it is

necessary to know the relative abundance of each neutral particle species as well as the neutral particle polarizability which can be found in (e.g., Schunk and Nagy, 2009). Since this method evaluates the collisions for each ion and neutral species separately, resonant ion-neutral interactions between the same species have to be considered. The parameters for resonant collisions are available in Schunk and Nagy (2009) as well. Resonant collision parameters depend on the ion temperature $T_i$ which is available from the EISCAT measurements.

Figure 6 shows two neutral particle density profiles calculated with different collision models compared to two profiles directly taken from neutral atmosphere models. The neutral densities are calculated from the ion-neutral collision frequency profile obtained for the 29 August 2013 campaign with the $\beta$ parameter determined at 150 km altitude. The uncertainties shown in Figure 6 are predominantly caused by the uncertainties of the collision frequency profile in Figure 3. The geophysical variation of the model neutral atmosphere background is comparably small and therefore the uncertainties are the same for

all measured profiles. The profile calculated from the rigid-sphere collision formula in Eq. 1 assumes the mean particle mass $A$ as given by the GAIA model which spans a range of $\sim 26 - 29$ amu across the transition region. Neutral density profiles calculated from Eq. 1 for the $A$ profile from the NRLMSIS 2.0 model or a constant $A = 30.5$ amu profile, which is a previously used assumption for the transition region (e.g. Nozawa et al., 2010), are nearly identical to the one shown in Fig. 6 and are therefore not shown as well. The neutral densities calculated under the assumption of Maxwell collisions (Schunk and Nagy,

2009) are nearly equivalent to those calculated for rigid-sphere collisions. The Maxwell-collision neutral density profile, too, is only shown for the GAIA model atmospheric composition since profiles calculated for different compositions are nearly identical. It can be seen that the neutral density profiles calculated from ion-neutral collision measurements are not sensitive to either the choice of collision model nor the assumed atmospheric composition. Furthermore, we added two neutral density profiles from atmosphere models for comparison. The NRLMSIS 2.0 profile is the same one used to calculate the ion-neutral

collision frequency profiles in Fig. 3. The neutral density profile obtained from the GAIA model shows a slightly different values but also without any vertical structure of the neutral density other than the smooth climatology.

The variability of the measured neutral density profile $n_{n,mes}$ can be seen from the relative variation $(n_{n,mes} - n_{n,MSIS})/n_{n,MSIS} = \delta n_n/n_{n,MSIS}$. For better comparison with Nicolls et al. (2014), we calculate the relative variation of neutral mass density $\delta\rho/\rho_{MSIS}$. Since the atmospheric composition has to be assumed for the calculation of neutral particles already, this step does

not require any additional assumptions.

Figure 7 shows the relative variation of measured neutral mass density compared to the NRLMSIS 2.0 model. It can be seen that the neutral mass density measurements oscillate around the MSIS climatology below 120 km altitude with a relative amplitude of $\sim 0.5 - 1$. The obtained profile renders previously presented results (Fig. 5 in Nicolls et al., 2014). Deviations of the vertical profile from the climatological background were interpreted as tidal or lower frequency oscillations.



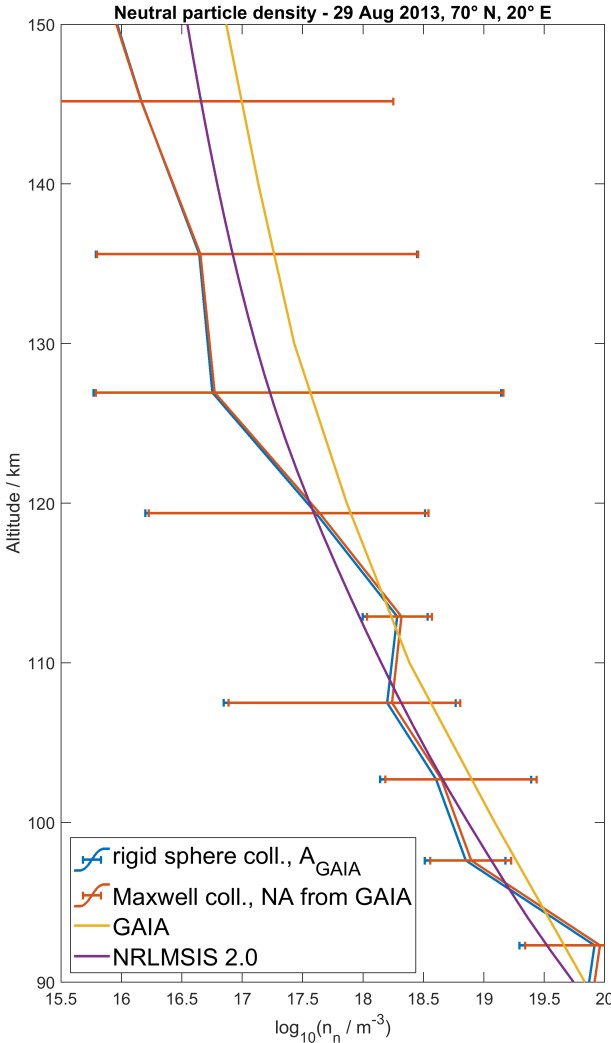

**Figure 6.** Neutral density profiles calculated from the measured ion-neutral collision frequencies for either rigid-sphere or Maxwell collisions. For both methods, the mean neutral mass and the abundances of the different neutral species $NA$ have been taken from the GAIA model. For comparison, profiles from the NRLMSIS 2.0 and GAIA models are shown.

## 5 Discussion

Since there have been no previous experimental studies applying the *difference spectrum* method, the obtained results can only be compared to multifrequency measurements applying a different analysis method and both direct and indirect measurements of the ion-neutral collision frequency. The $\nu_{in}$ profiles in Fig. 3 can be compared to the results by Nicolls et al. (2014) since they were obtained from the same measurements. The profile obtained for the $\beta$ parameter determined at $h_\beta = 150$ km altitude agrees well with the results obtained in Nicolls et al. (2014). As shown in Fig. 4, the $\beta$ parameter is altitude dependent for



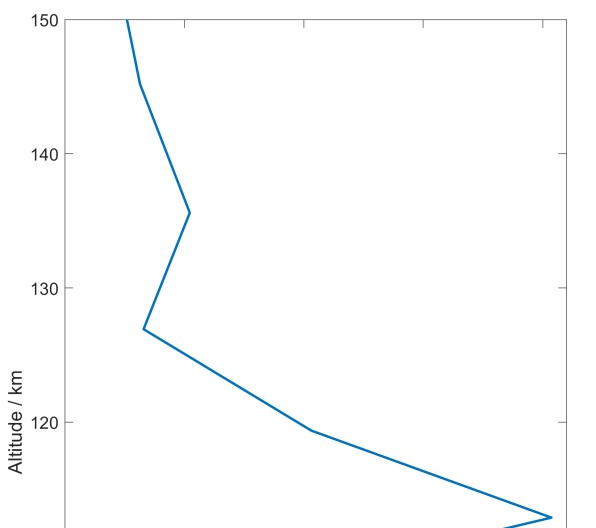

**Figure 7.** Variation of the neutral mass density calculated from the measured ion-neutral collision frequency profile relative to the NRLMSIS 2.0 mass density.

the EISCAT campaign on 29 August 2013, though it is expected to be roughly constant for all altitudes where $\nu_{in} \approx 0$ can be assumed (Grassmann, 1993b). The $\beta$ profile for the EISCAT measurements from 27 September 2021 shows a distinctly lower rate of change with altitude. Since there are more than eight years between the two measurements, technical updates of the system might explain the different behavior of the parameter. Since the beam shapes of the UHF and the VHF system are not identical, the scatter volumes are close but also not identical. Therefore, already minor system updates or degradations of one of the systems can significantly impact the result of our dual-frequency analysis which is performed under the assumption of identical scatter volumes. The decrease of $\beta$ with altitude indicates, that the amplitude of the UHF spectrum decreases stronger with altitude than the amplitude of the VHF spectrum.

The $\nu_{in}$ profile obtained for 27 September 2021, shown in Fig. 2 shows an increased collision frequency above 115 km altitude compared to the climatology. This agrees with previous findings (Fig. 2 in Nygrén, 1996) which show an increase of $\nu_{in}$



compared to the MSIS-86 climatology above 110 km for a single campaign on 26 August 1985. Nygrén (1996) combined both the direct measurement of $\nu_{in}$ from the ISR spectrum assuming $T_e = T_i$ and the indirect measurement from vertical ion drifts. Oyama et al. (2012) also reported an increased collision frequency above about 120 km during ionospheric heating events at E-region altitudes. They interpreted this as the result of an upward motion of denser neutral gas from lower altitudes. In

Fig. 2, the minimum of $\nu_{in}$ around 105 km altitude in combination with the very slow decrease above could be explained by thermospheric gas being transported upward from the altitude of the minimum.

Ion-neutral collision frequency measurements are of special interest for atmospheric physics since they allow inferring information about neutral gas densities in the MLT region. There are multiple methods to calculate collision frequencies from neutral particle densities, two of which are presented in Fig. 6 for the EISCAT measurements on 29 August 2013. One method as-

sumes both ions and neutrals to be rigid spheres while the other assumes Maxwell collision between the ions and the polarized neutrals. While the rigid sphere collision model only requires assumptions on the mean neutral particle mass, the abundances of the different neutral particle sorts have to be assumed for the Maxwell collision model. Figure 6 showed that the choice of the collision model has a far greater impact on the neutral density profile than the assumptions about mean neutral mass or particle abundances. Comparison to one empirical atmosphere model (NRLMSIS 2.0) and one physics-based model (GAIA)

resulted in expectable agreement and disagreement. Both models display a smooth neutral density profile and do not capture small-scale dynamics, but on the other hand, indicate the expected vertical behavior. For validation of our measurements, a gravity wave resolving model which includes incompressibility terms for small scales such as the High Altitude Mechanistic general Circulation Model (HIAMCM) (Becker and Vadas, 2020) would be required. In future studies, the inferred neutral density profiles could also be validated by comparison to meteor radar measurements. The neutral particle density can be obtained

from meteor radar measurements with the meteor peak flux altitude as a proxy (Stober et al., 2012). Figure 7 was designed following the example of (Fig. 5 in Nicolls et al., 2014) and agrees reasonably well. The relative variation of the neutral mass density shows periodic oscillations which were interpreted as the result of tides and lower frequency oscillations by Nicolls et al. (2014).

Additionally, there are possible improvements to the general analysis of ISR measurements, applicable for all described colli-

sion frequency measurements, including the one presented in this paper. One possible improvement for all discussed analysis techniques could be the application of full-profile analysis (Lehtinen et al., 1996). Instead of analyzing each altitude gate independently, as is done in this study and Nicolls et al. (2014), the total vertical profile of the plasma parameters could be fitted during full-profile analysis. The assumption that the plasma parameters are constant within each altitude gate is therefore not required for full-profile analysis (Lehtinen et al., 1996).

**6 Conclusions**

In this paper, we presented the first application of the *difference spectrum* method to analyze multifrequency ISR measurements to obtain direct measurements of the ion-neutral collision frequency. We have shown that this method can be applied in combination with standard ISR analysis software (GUISDAP). Comparison to the only previous multifrequency ISR measurement,




which applied a special software to analyze two ISR measurements simultaneously, showed reasonable agreement. Therefore
the *difference spectrum* method can be applied as an equivalent multifrequency ISR analysis method, generally applicable
without the requirement of a highly specialized ISR analysis software. This is the main advantage of the method presented here
over other multifrequency methods. Multifrequency methods are generally advantageous for collision frequency measurements
since they do not require additional assumptions like $T_e = T_i$ or strong electric fields.

We presented ion-neutral collision frequency profiles from three different EISCAT multifrequency campaigns. The measurements from 29 August 2013 were applied for comparison to the other multifrequency analysis method. Contrary to expectation,
the scaling parameter $\beta$ was not constant at nearly collisionless altitudes which could indicate a problem with one of the systems at the time. The second multifrequency campaign that we analyzed was conducted on 27 September 2021 in the same
radar mode as the previous campaign. The vertical $\beta$ parameter profile exhibited a significantly lower gradient in the F region
for these measurements. At about $100 - 115$ km altitude, the measurements showed a notably lower collision frequency than
the climatology, while the collision frequency was larger than the climatology above 115 km. This might be the result of an
upward motion of neutral gas due to ionospheric heating, e.g. caused by Joule dissipation. The third analyzed multifrequency
campaign was conducted on 13 October 2022 and applied a different radar mode than the previous ones. The EISCAT *manda*
radar mode allows measurements in the lower thermosphere with a very high altitude resolution which might be helpful to
study phenomena with a small-scale altitude structure.

In general, further improvement of the *difference spectrum* method as well as multifrequency experiments is required. Possible
improvements would be the already mentioned application of full-profile analysis of the ion-neutral collision frequency or the
general improvement of ISR fitting by including the exact ion chemistry of the ionosphere. Validation of our collision frequency
and neutral density measurements is difficult due to the general lack of observational methods in the ionospheric transition region. However, the *manda* measurements cover and resolve a good part of the meteor radar altitudes at about $80 - 110$ km.
Meteor radar neutral density measurements could be one possibility to verify multifrequency ISR experiments. As shown in
Section 4, the comparison to neutral atmosphere models is only somewhat meaningful, as long as the models do not include the
incompressibility terms for small-scale variability e.g., due to gravity waves. Considering the lack of atmospheric measurement
methods in the ionospheric dynamo region, the method presented here is highly valuable because it is providing information
about this important region.

*Data availability.* The data are available under the Creative Commons Attribution 4.0 International license at https://doi.org/10.5281/zenodo.8074787.

*Author contributions.* FG performed the data analysis and wrote large parts of the manuscript. DP suggested the idea to analyze multifrequency experiments. All authors provided feedback and were involved in revising the manuscript.



*Competing interests.* We declare no competing interests.

*Acknowledgements.* EISCAT is an international association supported by research organizations in China (CRIRP), Finland (SA), Japan
(NIPR and ISEE), Norway (NFR), Sweden (VR), and the United Kingdom (UKRI). The dataset used for this study is from the Ground-to-topside model of the Atmosphere and Ionosphere for Aeronomy (GAIA) project carried out by the National Institute of Information and Communications Technology (NICT), Kyushu University, and Seikei University. Gunter Stober is a member of the Oeschger Center for Climate Change Research (OCCR).





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
