# Peer review of "Difference spectrum fitting of the ion-neutral collision frequency from dual-frequency EISCAT measurements"

_EGUsphere, 2023_

## Author Response (AR1)

**Changes made to the manuscript:**

We added two brief statements to the manuscript that hopefully cover the questions asked by Referee 2.

**Referee 1:**

We thank the Referee for taking the time to read our manuscript and for their positive feedback.

**Referee 2:**

We thank the Referee for taking the time to review our paper. Please find the responses to your question below.

**Referee:**

As you mentioned in the paper, you compare the difference of the scaling parameter β depending on the altitude. Do you analyze its change with time? What is the suitable strategy to choose the scaling parameter β?

**Authors:**

We determine the β parameter profile for each 60s integration window separately. The β profiles shown in Figure 4 are the median profiles of the respective campaigns. The β value does not show a significant trend over the course of one campaign (few hours) at any altitude. Since the β parameter is introduced to account for technical differences between the UHF and VHF systems, changes within a few hours are not expected. However, there are distinct outliers for some integration windows, presumably during which one of the instruments failed to measure a clear ISR spectrum allowing for analysis. Therefore, median statistics was chosen as the appropriate strategy to determine the scaling parameter β.

**Referee:**

If the frequency of two ISRs is close. Does the frequency difference of ISR effect the measurements?

**Authors:**

The important parameter here is not the difference of radar frequencies but their ratio ξ. As described in Equation 3, the simultaneous UHF and VHF measurements are similar to two UHF measurements at $\nu_{in}$ and $\xi \cdot \nu_{in}$. This causes the difference of the two spectra. For a ξ ratio close to unity, the difference spectrum is extremely weak and overshadowed by measurement uncertainties. Inferring the ion-neutral collision frequency is therefore only possible for an ξ ratio distinctly larger than 1 (4.2 for the EISCAT systems).